# Synthesis of 3,5-Bis(trifluoromethyl)phenyl-Substituted Pyrazole Derivatives as Potent Growth Inhibitors of Drug-Resistant Bacteria

**DOI:** 10.3390/molecules26165083

**Published:** 2021-08-22

**Authors:** Ibrahim S. Alkhaibari, Hansa Raj KC, Subrata Roy, Mohd. K. Abu-gazleh, David F. Gilmore, Mohammad A. Alam

**Affiliations:** 1Department of Chemistry and Physics, The College of Sciences and Mathematics, Arkansas State University, Jonesboro, AR 72401, USA; ibrahimsaleh.s@hotmail.com (I.S.A.); hansa.kc@smail.astate.edu (H.R.K.); subrata.roy@smail.astate.edu (S.R.); mohdkota.abugazle@smail.astate.edu (M.K.A.-g.); 2Department of Biological Sciences, The College of Sciences and Mathematics, Arkansas State University, Jonesboro, AR 72401, USA; dgilmore@astate.edu

**Keywords:** pyrazole, aniline, *Staphylococcus aureus*, *Enterococcus faecium*, MRSA, *Enterococcus faecalis*, VRE

## Abstract

Enterococci and methicillin-resistant *S. aureus* (MRSA) are among the menacing bacterial pathogens. Novel antibiotics are urgently needed to tackle these antibiotic-resistant bacterial infections. This article reports the design, synthesis, and antimicrobial studies of 30 novel pyrazole derivatives. Most of the synthesized compounds are potent growth inhibitors of planktonic Gram-positive bacteria with minimum inhibitory concertation (MIC) values as low as 0.25 µg/mL. Further studies led to the discovery of several lead compounds, which are bactericidal and potent against MRSA persisters. Compounds **11**, **28**, and **29** are potent against *S. aureus* biofilms with minimum biofilm eradication concentration (MBEC) values as low as 1 µg/mL.

## 1. Introduction

Drug-resistant ESKAPE (*Enterococcus faecium*, *Staphylococcus aureus*, *Klebsiella pneumoniae*, *Acinetobacter baumannii*, *Pseudomonas aeruginosa*, and *Enterobacter* species) pathogens cause a majority of nosocomial infections all over the world. These multidrug-resistant bacteria have reduced treatment options, increased hospital stays and treatment costs, and amplified the death rate of infected patients [1]. *E. faecium* causes a variety of problems, including urinary tract, intra-abdominal, pelvic, and soft tissue infections, bacteremia, and endocarditis [2]. Approximately 30% of all healthcare-associated enterococcal infections are caused by *E. faecalis* and *E. faecium*, which are vancomycin-resistant (VRE), and these resistant strains are increasingly becoming resistant to other antibiotics. VRE is the most common cause of central line-associated bloodstream infections [3]. Furthermore, enterococcal biofilms cause 25% of all catheter-associated urinary tract infections [4]. Alcohol tolerance (ethanol and isopropanol) of clinical isolates of *E. faecium* has increased over the years, and isolates obtained after 2010 are 10-fold more tolerant to killing by alcohol than were older isolates [5]. *S. aureus* infections are caused by different strains, including methicillin-sensitive *S. aureus* (MSSA), methicillin-resistant *S. aureus* (MRSA), vancomycin-intermediate *S. aureus* (VISA), and vancomycin-resistant *S. aureus* (VRSA). Infections by MRSA are the most problematic, but any *S. aureus* infection can become serious [6]. MRSA causes ten-fold more infections than all multidrug-resistant (MDR) Gram-negative bacteria combined [7]. MRSA has emerged as one of the most menacing pathogens of humans, and this pathogen currently bypasses HIV in terms of fatality rate [8]. Hospitals are the hotbeds for highly drug-resistant pathogens, such as MRSA, increasing the risk of hospitalization kills instead of cures [9].

1*H*-Pyrazole (1,2-diazole) is a five-membered heterocycle, and its derivatives are known for a wide spectrum of biological activities [10]. It is found as the core structure of several leading drugs, such as celecoxib, a potent anti-inflammatory medicine [11]; tepoxalin, a nonsteroidal anti-inflammatory agent for veterinary use [10]; the anti-obesity drug, rimonabant [12]; the analgesic difenamizole [13]; and several other therapeutic agents. A number of drugs containing pyrazole nuclei may be due to its decreased susceptibility to oxidative degradation metabolism compared to other five-membered heterocycles [14]. Pyrazole derivatives have been reported as antimicrobial agents in several publications [15,16,17]. Although pyrazole derivatives as anti-MRSA agents have been reported by us [18,19,20,21,22] and others [23,24], pyrazole derivatives with anti-Acinetobacterial activity had not been reported until our recent papers [18,19,20,21,22].

The trifluoromethyl (–CF_3_) group strategically placed on a phenyl ring is known to improve the pharmacodynamics and pharmacokinetic properties of the resulting compounds [25]. A number of widely used drugs, such as dutasteride [26], hydroxyflutamide [27], and cinacalcet [28], contain the trifluoromethyl substituted phenyl moiety (Figure 1). We have found the trifluoromethyl substituted phenyl groups as potent growth inhibitors of different bacterial strains, including MRSA [22,29]. Based on the significance of the trifluoromethyl group in drug discovery, herein, we report the synthesis and antimicrobial activities of 3,5-bis(trifluoromethyl)phenyl substituted pyrazole derivatives as potent antimicrobial agents.

## 2. Results and Discussion

The designed compounds were synthesized by the reaction of 3′,5′-bis(trifluoromethyl)acetophenone (**I**) with 4-hydrazinobenzoic acid (**II**) to form the hydrazone intermediate, which on reaction with the Vilsmeier-Haack reagent formed the pyrazole aldehyde (**III**) [21,29,30,31]. This aldehyde intermediate (**III**) was synthesized on a 10 g scale, and the pure product was obtained simply by filtration and washing with water. Reductive amination of the aldehyde (**III**) with different anilines (R-NH_2_) formed the target compounds (**1**–**30**) in a very good average yield (Table 1). The synthetic method is very robust as different substituents on the aniline moiety did not affect the yield and purity of the products. 

All the synthesized compounds were tested for their activity against Gram-positive and Gram-negative bacteria. These compounds showed broad activity against Gram-positive strains, but no activity against the Gram-negative strains (Table 2). The phenyl-substituted derivative (**1**) showed moderate activity against the tested bacteria with minimum inhibitory concentration (MIC) values as low as 2 µg/mL. Alkyl substitution on the phenyl ring of the aniline moiety increased the activity of the resultant compounds (**2** and **3**). 4-*Iso*propyl aniline derivative (**2**) inhibited the growth of *S. aureus* strains with MIC values in the range of 1 to 2 µg/mL. Similar potency was observed against *Staphylococcus epidermidis*, enterococci, and *Bacillus subtilis* strains. Methoxy substituent decreased the potency of the product (**4**) significantly. Activities of the compounds so far indicated that hydrophobic substituents on the phenyl ring increased the activity of the compounds. The phenoxy-substituted derivative was found to be a potent antimicrobial compound with MIC values of 1 µg/mL against several bacterial strains. The methyl sulfide attached compound (**6**) showed activity with MIC values in the range of 1 to 4 µg/mL. Fluorine substituted compounds (**7** and **8**) were good bacterial growth inhibitors. 3-Fluoro derivative (**7**) was found to be slightly less potent than the 4-fluoro derivative (**8**). Chloro-substitution resulted in a compound (**9**) with better activity than the fluoro derivatives. Bromo derivatives (**10** and **11**) are both potent antimicrobial agents, and the 4-chloro (**9**) showed similar antimicrobial activity. The carboxylic acid functional group eliminated the activity of the resultant compounds (**12** and **13**).

We found the disubstituted aniline derivatives as potent antibacterial compounds. Both methyl and halogen-substituted compounds (**14**, **15**, and **16**) showed MIC values as low as 1 µg/mL. 4-Bromo-3-methyl derivative (**16**) appeared the most potent compound among these three derivatives. Tested bacteria were less susceptible to the difluorophenyl-substituted compound (**17**) with MIC values 1–4 µg/mL across the tested strains. Chloro-fluoro substitution (**18**) showed better activity than the difluoro derivative (**17**). Dichloro aniline derivatives (**19** and **20**) were among the most potent compounds in the series, with MIC values as low as 0.5 µg/mL. 4-Bromo-3-chloro-aniline-substituted pyrazole derivative (**21**) was very effective with MIC values as low as 0.5 μg/mL against *S. aureus*, including MRSA strains. Tested enterococci strains were also inhibited efficiently by this compound (**21**) with the MIC values of 1 μg/mL. Trifluoromethyl derivatives (**22** and **23**) showed good activity across the tested strains. Trifluoro aniline derivatives (**24** and **25**) were comparatively less potent than other trisubstituted derivatives. All the mixed trihalo derivatives (**26**, **27**, **28**, and **29**) showed good potency. Compound **26** inhibited the growth of many strains with MIC values as low as 0.25 μg/mL. The last compound (**30**) in the series was a good inhibitor of tested bacterial strains. The potency of most of the compounds was shown to be better than that of vancomycin and daptomycin, two widely used antibiotics to treat Gram-positive bacterial infections. Several compounds (e.g., **19**, **20**, **21**, **22**, **26**, etc.) were up to 4 times more effective than these two positive controls against *S. aureus* strains. Some compounds (e.g., **3** and **19**) were more potent than vancomycin and daptomycin against *S. epidermidis* bacterium. Most of the compounds were very potent against the enterococci strains compared to the positive controls. 

Based on the activities, we found a good Structure Activity Relationship (SAR) of the product with respect to the substituents in the aniline moiety. The presence of a protic substituent, carboxylic acid, eliminated the activity of the compounds (**12** and **13**), which is in accordance with our previous findings [29,31,32]. The presence of lipophilic substituents (clog P = 7–8) on the phenyl ring of the aniline moiety increased the activity. As can be seen from the activity of mono-substituted compounds (**2**–**11**), the methoxy substituent is the least lipophilic (clog P = 6.63), and the corresponding compound (**4**) is the least potent for its antibacterial activity. *Para* isomers (**8** and **11**) are more potent than the corresponding *meta* products (**7** and **10**). Similar SAR was observed for disubstituted derivatives (**14**–**23**). Trisubstituted (**24**–**29**) and tetrasubstituted (**30**) compounds showed similar average activity as the disubstituted compounds.

### 2.1. Cytotoxicity Studies

After determining a high level of potency for several compounds, we tested them for their possible toxicity to cultured human cells (HEK-293) to select the least toxic compounds for further studies. Initially, compounds were tested at 50 and 25 µg/mL. Unfortunately, several compounds showed a toxic effect on HEK-293 cells. Compounds allowing >20% viability of HEK-293 cells at the tested concentrations were selected to test their toxicity at lower concentrations by determining their IC_50_ values (Figure 2). Compound **5** was the least toxic with an IC_50_ value of 9.15 µg/mL. Compounds **11**, **28**, and **29** inhibited the growth of these cells with IC_50_ values in the range of 7–8 µg/mL.

### 2.2. Time Kill Assay

Time Kill Assay studies give an idea of whether antibacterial agents are bactericidal or bacteriostatic. As can be seen in Figure 3, all four compounds were bactericidal, reducing bacterial concentrations 99.9% by 4 h and eliminating viable cells by 6 to 8 h. The positive controls, vancomycin (**V**) and daptomycin (**D**), eliminated >99.9% CFUs in 8 and 6 h, respectively. The negative control, DMSO, showed exponential growth as expected.

### 2.3. Activity against Persisters

Persisters are a nongrowing, antibiotic-tolerant state of bacteria. MRSA persisters are one of the major contributors to its virulence and lethality. Recently, a number of efforts are being made to eliminate MRSA persisters by using novel antibiotics [33,34,35]. In our effort to get potent anti-MRSA agents, we tested our promising compounds for their potency against persisters. Three of the four potent compounds (**5**, **11**, **28**, and **29**) are very effective against *S. aureus* ATCC 700699 persisters. Compound **5** showed partial activity against the persisters (Figure 4). Compounds **11** and **29** were very effective at eliminating the persisters at 32× MIC value. Compared to the positive controls, these two compounds reduced the population of persisters by 4 log values. Compound **28** is also very effective against the persisters, but less potent than of compounds **11** and **29**. Conventional antibiotics, vancomycin and daptomycin, failed to demonstrate persister kill activity at 32× MIC. Compounds **11** and **29** were further studied to examine the time-kill pattern during an 8 h period at various MIC concentrations (Figure 4b,c). Both compounds **11** and **29** eradicated more than 3 log values within an 8 h time course at all the 4× MIC, 8× MIC, and 16× MIC concentrations, demonstrating excellent bactericidal activity against MRSA persisters. The 2× MIC concentration of compound **11** eliminated 1 log value of the cells, while 2× MIC of compound **29** demonstrated almost 2 log values decrease of persister cells in 8 h time period. The level of detection in this experiment was 2 × 10^4^ CFU/mL, and the data point on log_10_ 4 in the graph was below the level of detection (Figure 4a–c).

### 2.4. Activity against S. aureus and E. faecalis Biofilms

Microbial biofilms are self-synthesized polymeric matrices that help microbial populations adhere to abiotic and biotic surfaces. Biofilms make bacteria recalcitrant to antibiotic treatment and protection from the host immune system. Biofilms formed by staphylococci (mostly by *S. epidermidis* and *S. aureus*) are the most common cause of device-related infections in hospital settings [7]. Enterococci biofilms are also frequently observed in the dysbiotic gastrointestinal tract, endocarditis, and wounds [4].

We tested the compounds showing potent activity against planktonic bacterial strains for their ability to eradicate the *S. aureus* and *E. faecalis* biofilms by using the Calgary Biofilm Device. As shown in Table 3, compound **11** showed potent antibiofilm property with the MBEC values 2 and 4 µg/mL against *S. aureus* and *E. faecalis*, respectively. Compounds **28** and **29** are even better with MBEC values that are as low as 1 µg/mL. Compound **5** did not show any appreciable biofilm eradication property. Comparing with the positive control, vancomycin (**V**) and daptomycin (**D**), our compounds (**11**, **28**, and **29**) are very potent in eradicating the *S. aureus* and *E. faecalis* biofilms. 

## 3. Procedures

### 3.1. General Consideration

All of the reactions were carried out under an air atmosphere in round-bottom flasks. All reactant materials, solvents, deuterated solvents for ^1^H and ^13^C Nuclear Magnetic Resonance (NMR) spectroscopy, and for the reactions were purchased from Fisher Scientific (Hanover Park, IL, USA) and Oakwood chemical (Estill, SC, USA). ^1^H and ^13^C NMR spectra were recorded on a Varian Mercury with 300 MHz for ^1^H and 75 MHz for ^13^C NMR. The Fourier Transform Infrared (FTIR) spectra were obtained using 7 mm KBr pellets for the specimen using a Nicolet™ iS™ 10 FTIR spectrometer (Thermo Fisher Scientific Inc., Waltham, MA, USA). High Resolution Mass Spectrometry (HRMS) data were obtained by using the Brucker Apex II-FTMS system.

### 3.2. Synthesis of 3,5-Bis(trifluoromethyl)phenyl-derived Pyrazole Aldehyde (**III**)

4-Hydrazinobenzoic acid (1.597 g, 10.5 mmol) and 3,5-bis(trifluoromethyl)acetophenone (2.561 g, 10 mmol) were taken in a 100 mL round-bottom flask. Anhydrous ethanol (50 mL) was added to the flask, and the reaction mixture was refluxed for 8 h. After completing the reaction, the solvent was removed under reduced pressure using a rotary evaporator, resulting in a dry product. Then 30 mL anhydrous *N*,*N*-dimethylformamide (DMF) was added to the flask and sealed with a septum followed by stirring for 15 min to dissolve the compound. After getting a clear solution, the flask was cooled using an ice bath. Then phosphorous oxychloride, POCl_3_ (4.67 mL, 50 mmol) was added dropwise using a syringe through the septum. After 30 min, the mixture was brought to ambient temperature and heated at 90 °C for 8 h. Then, the reaction mixture was poured into an ice-full beaker (300 mL) and stirred for 10 h to make precipitate. The precipitate was filtered and washed with water several times. Drying under vacuum gave the final pure 3,5-bis(trifluoromethyl)phenyl-derived pyrazole aldehyde (**III**).

### 3.3. Synthesis of 3,5-Bis(trifluoromethyl)phenyl Substituted Pyrazole-Derived Anilines (**1**–**30**)

The pyrazole aldehyde (214.14 mg, 0.5 mmol) and 0.55 mmol aniline derivative were taken in a round-bottom flask and refluxed in toluene for 6 h in a Dean-Stark condenser. Then, the reaction mixture was cooled and filtered under vacuum. The precipitate was recrystallized in acetonitrile to give a pure imine product. The imine product (0.5 mmol) was dissolved in methanol and cooled to 0 °C using an ice bath. NaBH_4_ (94.5 mg, 2.5 mmol) was added to the solution before stirring the mixture for 10 h. After that 10% HCl was added to form the precipitate. The precipitate was recrystallized in acetonitrile to obtain the pure 3,5-bis(trifluoromethyl)phenyl substituted pyrazole-derived aniline product.

### 3.4. Compound Characterization Data



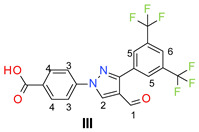



*4-[3-[3,5-Bis(trifluoromethyl)phenyl]-4-formyl-pyrazol-1-yl] benzoic acid* (**III**). Beige solid; (420 mg, 98%); IR (KBr pellet, cm^−1^): 3095 (aromatic C-H), 2672, 1694 (C=O), 1608, 1131 (C-F); ^1^H NMR, 300 MHz (DMSO-*d*_6_): δ 9.96 (s, 1H, H-1), 9.58 (s, 1H, H-2), 8.70 (s, 2H, H-4), 8.21–8.11 (m, 5H, H-3, H-5 and H-6);^13^C NMR (75 MHz, DMSO-d_6_): δ 184.8, 166.9, 162.7, 149.3, 141.6, 138.7, 133.9 (^3^*J*_C-F_ = 4.0 Hz), 131.4, 130.8 (^2^*J*_C-F_ = 33.2 Hz), 129.4, 129.1, 123.7 (^1^*J*_C-F_ = 270.6 Hz), 123.2, 119.5 (m). HRMS (ESI-FTMS Mass (*m*/*z*): calcd for C_19_H_10_F_6_N_2_O_3_ [M + H]^+^ = 429.0668, found 429.0747. (Appendix A)



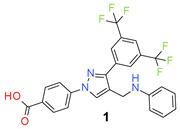



*4-[4-(Anilinomethyl)-3-[3,5-bis(trifluoromethyl)phenyl]pyrazol-1-yl] benzoic acid* (**1**). Light beige solid; (436 mg, 86%); IR (KBr pellet, cm^−1^): 3415, 2894, 1699, 1608, 1360, 1316, 1278, 1174, 1134; ^1^H NMR, 300 MHz (DMSO-*d*_6_): δ 8.92 (s, 1H), 8.22 (s, 2H), 8.09–7.98 (m, 5H), 7.15 (d, *J* = 7.4 Hz, 2H), 7.00–6.92 (m, 3H), 4.45 (s, 2H); ^13^C NMR (75 MHz, DMSO-d_6_): δ 167.0, 149.8, 142.3, 135.0, 132.0, 131.5, 130.8 (^2^*J*_C-F_ = 32.9 Hz), 130.3, 129.6, 129.3, 129.0, 128.5, 123.6 (^1^*J*_C-F_ = 270.2 Hz), 122.1, 118.6, 116.4, 41.9. HRMS (ESI-FTMS Mass (*m*/*z*): calcd for C_25_H_17_F_6_N_3_O_2_ [M + H]^+^ = 506.1298, found 506.1291.



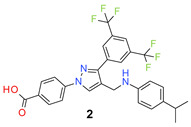



*4-[3-[3,5-Bis(trifluoromethyl)phenyl]-4-[(4-isopropylanilino)methyl]pyrazol-1-yl] benzoic acid* (**2**). Light beige solid; (443 mg, 80%); IR (KBr pellet, cm^−1^): 3411, 2969, 1714, 1607, 1358, 1281, 1172, 1127; ^1^H NMR, 300 MHz (DMSO-*d*_6_): δ 8.92 (s, 1H), 8.31 (s, 2H), 8.06 (t, *J* = 8.7 Hz, 5H), 7.08 (d, *J* = 8.0 Hz, 2H), 6.94 (br, 2H), 4.48 (s, 2H), 2.75–2.73 (m, 1H), 1.11 (d, *J* = 6.7 Hz, 6H); ^13^C NMR (75 MHz, DMSO-d_6_): δ 167.0, 149.5, 142.3, 135.1, 132.1, 131.6, 131.0 (^2^*J*_C-F_ = 32.8 Hz), 130.4, 129.3, 129.0, 128.4, 127.4, 123.6 (^1^*J*_C-F_ = 271.2 Hz), 122.1, 118.6, 118.2, 43.0, 33.1, 24.1. HRMS (ESI-FTMS Mass (*m*/*z*): calcd for C_28_H_23_F_6_N_3_O_2_ [M + H]^+^ = 548.1767, found 548.1761.



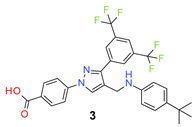



*4-[3-[3,5-Bis(trifluoromethyl)phenyl]-4-[(4-tert-butylanilino)methyl]pyrazol-1-yl] benzoic acid* (**3**). White solid; (526 mg, 93%); IR (KBr pellet, cm^−1^): 3419, 2969, 1712, 1607, 1357, 1281, 1178, 1131; ^1^H NMR, 300 MHz (DMSO-*d*_6_): δ 8.92 (s, 1H), 8.32 (s, 2H), 8.06 (t, *J* = 8.4 Hz, 5H), 7.23 (d, *J* = 8.0 Hz, 2H), 6.95 (d, *J* = 7.3 Hz, 2H), 4.49 (s, 2H), 1.18 (s, 9H); ^13^C NMR (75 MHz, DMSO-d_6_): δ 167.0, 149.5, 142.3, 135.1, 132.2, 131.6, 131.2, 130.8 (^2^*J*_C-F_ = 32.6 Hz), 130.4, 129.4, 128.3, 126.4, 123.6 (^1^*J*_C-F_ = 271.2 Hz), 122.2, 118.6, 118.2, 43.0, 34.4, 31.4. HRMS (ESI-FTMS Mass (*m*/*z*): calcd for C_29_H_25_F_6_N_3_O_2_ [M + H]^+^ = 562.1924, found 562.1919.



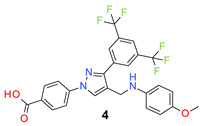



*4-[3-[3,5-Bis(trifluoromethyl)phenyl]-4-[(4-methoxyanilino)methyl]pyrazol-1-yl] benzoic acid* (**4**). Light grey solid; (417 mg, 77%); IR (KBr pellet, cm^−1^): 3219, 2261, 1701, 1608, 1280, 1074, 1032; ^1^H NMR, 300 MHz (DMSO-*d*_6_): δ 9.03 (s, 1H), 8.09 (t, *J* = 9.3 Hz, 5H), 7.99 (d, *J* = 8.4 Hz, 2H), 7.11 (d, *J* = 8.4 Hz, 2H), 6.72 (d, *J* = 8.5 Hz, 2H), 4.59 (s, 2H), 3.63 (s, 3H); ^13^C NMR (75 MHz, DMSO-d_6_): δ 166.9, 150.4, 142.2, 134.8, 133.1, 131.6, 130.6 (^2^*J*_C-F_ = 32.7 Hz), 129.5, 129.0, 128.7, 124.9, 123.6 (^1^*J*_C-F_ = 271.2 Hz), 118.7, 118.1, 115.2, 114.7, 55.5, 44.6. HRMS (ESI-FTMS Mass (*m*/*z*): calcd for C_26_H_19_F_6_N_3_O_3_ [M + H]^+^ = 536.1403, found 536.1397.



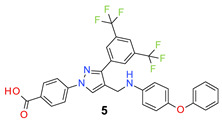



*4-[3-[3,5-Bis(trifluoromethyl)phenyl]-4-[(4-phenoxyanilino)methyl]pyrazol-1-yl] benzoic acid* (**5**). Off white solid; (477 mg, 79%); IR (KBr pellet, cm^−1^): 3416, 2749, 1692, 1607, 1313, 1179, 1140; ^1^H NMR, 300 MHz (DMSO-*d*_6_): δ 8.95 (s, 1H), 8.32 (s, 2H), 8.14–8.03 (m, 5H), 7.34 (t, *J* = 7.4 Hz, 2H), 7.11–7.01 (m, 3H), 6.88 (t, *J* = 7.6 Hz, 4H), 4.47 (s, 2H); ^13^C NMR (75 MHz, DMSO-d_6_): δ 167.0, 157.7, 149.6, 142.3, 135.1, 132.0, 131.5, 131.1 (^2^*J*_C-F_ = 32.7 Hz), 130.3, 129.3, 128.6, 128.4, 125.5, 123.6 (^1^*J*_C-F_ = 271.2 Hz), 123.4, 122.1, 120.3, 119.8, 119.4 118.6, 118.1. HRMS (ESI-FTMS Mass (*m*/*z*): calcd for C_31_H_21_F_6_N_3_O_3_ [M + H]^+^ = 598.1560, found 598.1556.



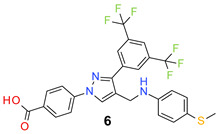



*4-[3-[3,5-Bis(trifluoromethyl)phenyl]-4-[(4-methylsulfanylanilino)methyl]pyrazol-1-yl] benzoic acid* (**6**). Light yellow solid; (389 mg, 70%); IR (KBr pellet, cm^−1^): 3412, 2924, 1701, 1608, 1359, 1281, 1177, 1131; ^1^H NMR, 300 MHz (DMSO-*d*_6_): δ 8.93 (s, 1H), 8.29 (s, 2H), 8.12–8.02 (m, 5H), 7.10 (d, *J* = 8.5 Hz, 2H), 6.91 (d, *J* = 8.3 Hz, 2H), 4.44 (s, 2H), 2.36 (s, 3H); ^13^C NMR (75 MHz, DMSO-d_6_): δ 167.0, 149.3, 142.4, 135.2, 131.5, 130.9 (^2^*J*_C-F_ = 32.8 Hz), 129.4, 129.2, 129.0, 128.2, 127.2, 124.2, 123.6 (^1^*J*_C-F_ = 271.3 Hz), 122.1, 119.5, 118.5, 118.2, 17.2. HRMS (ESI-FTMS Mass (*m*/*z*): calcd for C_26_H_19_F_6_N_3_O_2_S [M + H]^+^ = 552.1175, found 552.1173.



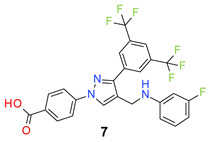



*4-[3-[3,5-Bis(trifluoromethyl)phenyl]-4-[(3-fluoroanilino)methyl]pyrazol-1-yl] benzoic acid* (**7**). White solid; (433 mg, 82%); IR (KBr pellet, cm^−1^): 3420, 2903, 1699, 1608, 1359, 1281, 1175, 1134; ^1^H NMR, 300 MHz (DMSO-*d*_6_): δ 8.84 (s, 1H), 8.40 (s, 2H), 8.07–8.11 (m, 5H), 7.10 (d, *J* = 7.05 Hz, 1H), 6.50 (t, *J* = 7.4 Hz, 2H), 6.38–6.29 (m, 1H), 4.29 (s, 2H); ^13^C NMR (75 MHz, DMSO-d_6_): δ 167.0, 163.7 (d, ^1^*J*_C-F_ = 238.4 Hz), 150.2 (d, ^3^*J*_C-F_ = 10.9 Hz), 148.9, 142.4, 135.3, 131.4 (^3^*J*_C-F_ = 4.6 Hz), 131.0, 130.7 (d, ^3^*J*_C-F_ = 10.2 Hz), 129.1, 127.9, 123.6 (^1^*J*_C-F_ = 271.2 Hz), 122.0, 119.9, 118.4, 109.5, 103.4 (d, ^2^*J*_C-F_ = 20.8 Hz), 99.4 (d, ^2^*J*_C-F_ = 24.8 Hz), 38.3. HRMS (ESI-FTMS Mass (*m*/*z*): calcd for C_25_H_16_F_7_N_3_O_2_ [M + H]^+^ = 524.1204, found 524.1208.



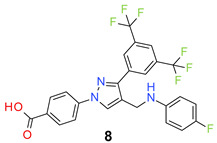



*4-[3-[3,5-Bis(trifluoromethyl)phenyl]-4-[(4-fluoroanilino)methyl]pyrazol-1-yl] benzoic acid* (**8**). Off white solid; (443 mg, 84%); IR (KBr pellet, cm^−1^): 3413, 2970, 1712, 1607, 1357, 1281, 1178, 1131; ^1^H NMR, 300 MHz (DMSO-*d*_6_): δ 8.92 (s, 1H), 8.26 (s, 2H), 8.09–8.04 (m, 5H), 6.99 (d, *J* = 6.7 Hz, 4H), 4.44 (s, 2H); ^13^C NMR (75 MHz, DMSO-d_6_): δ 167.0, 158.3 (d, ^1^*J*_C-F_ = 236.5 Hz), 149.7, 142.3, 138.8, 135.0, 132.0, 131.7, 131.0 (^2^*J*_C-F_ = 32.8 Hz), 129.3, 128.4, 125.4, 121.9 (d, ^2^*J*_C-F_ = 21.5 Hz), 121.4 (^1^*J*_C-F_ = 280.7 Hz), 118.6, 116.2 (d, ^2^*J*_C-F_ = 22.4 Hz), 41.6. HRMS (ESI-FTMS Mass (*m*/*z*): calcd for C_25_H_16_F_7_N_3_O_2_ [M + H]^+^ = 524.1204, found 524.1200.



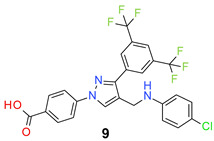



*4-[3-[3,5-Bis(trifluoromethyl)phenyl]-4-[(4-chloroanilino)methyl]pyrazol-1-yl] benzoic acid* (**9**). Yellowish white solid; (391 mg, 72%); IR (KBr pellet, cm^−1^): 3409, 2899, 1699, 1608, 1359, 1286, 1178, 1130; ^1^H NMR, 300 MHz (DMSO-*d*_6_): δ 8.84 (s, 1H), 8.39 (s, 2H), 8.12–8.07 (m, 5H), 7.13 (d, *J* = 8.4 Hz, 2H), 6.73 (d, *J* = 8.6 Hz, 2H), 4.29 (s, 2H); ^13^C NMR (75 MHz, DMSO-d_6_): δ167.0, 149.0, 146.5, 142.4, 135.3, 131.4 (^3^*J*_C-F_ = 7.1 Hz), 130.8 (^2^*J*_C-F_ = 42.8 Hz), 129.1 (m), 128.0, 123.6 (^1^*J*_C-F_ = 271.1 Hz), 123.2, 122.0, 121.3, 119.7, 118.5, 118.2, 115.1, 38.7. HRMS (ESI-FTMS Mass (*m*/*z*): calcd for C_25_H_16_ClF_6_N_3_O_2_ [M + H]^+^ = 540.0908, 542.0881, found 540.0907.



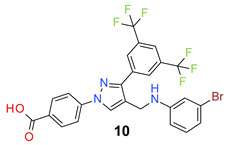



*4-[3-[3,5-Bis(trifluoromethyl)phenyl]-4-[(3-bromoanilino)methyl]pyrazol-1-yl] benzoic acid* (**10**). Off white solid; (497 mg, 85%); IR (KBr pellet, cm^−1^): 3410, 3126, 1706, 1607, 1373, 1277, 1188, 1137; ^1^H NMR, 300 MHz (DMSO-*d*_6_): δ 8.85 (s, 1H), 8.40 (s, 2H), 8.13–8.08 (m, 5H), 7.03 (t, *J* = 7.8 Hz, 1H), 6.86 (s, 1H), 6.73 (dd, *J* = 7.8, 14.5 Hz, 2H), 4.29 (s, 2H); ^13^C NMR (75 MHz, DMSO-d_6_): δ 167.0, 149.8, 148.9, 142.4, 135.3, 131.8, 131.5, 130.9, 130.5, 129.1, 127.9, 123.6 (^1^*J*_C-F_ = 271.1 Hz), 122.7, 122.0, 119.7 (^2^*J*_C-F_ = 26.2 Hz), 118.4, 115.2, 112.1, 38.1. HRMS (ESI-FTMS Mass (*m*/*z*): calcd for C_25_H_16_BrF_6_N_3_O_2_ [M + H]^+^ = 584.0403, 586.0384, found 584.0410.



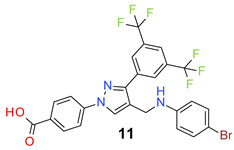



*4-[3-[3,5-Bis(trifluoromethyl)phenyl]-4-[(4-bromoanilino)methyl]pyrazol-1-yl] benzoic acid* (**11**). Yellow solid; (457 mg, 78%); IR (KBr pellet, cm^−1^): 3377,3072, 1712,1607, 1356,1282,1178, 1128; ^1^H NMR, 300 MHz (DMSO-*d*_6_): δ 8.83 (s, 1H), 8.40 (s, 2H), 8.12–8.04 (m, 5H), 7.24 (d, *J* = 8.6 Hz, 2H), 6.67 (d, *J* = 8.7 Hz, 2H), 4.27 (s, 2H); ^13^C NMR (75 MHz, DMSO-d_6_): δ 167.0, 149.0, 147.2, 142.4, 135.3, 131.6 (^2^*J*_C-F_ = 34.7 Hz), 131.0 (^3^*J*_C-F_ = 4.0 Hz), 130.5, 129.1, 127.9, 123.6 (^1^*J*_C-F_ = 271.2 Hz), 123.2, 122.0, 119.8, 118.4, 115.4, 108.4, 38.5. HRMS (ESI-FTMS Mass (*m*/*z*): calcd for C_25_H_16_BrF_6_N_3_O_2_ [M + H]^+^ = 584.0403, 586.0384, found 584.0408.



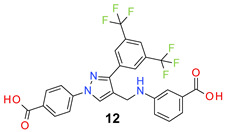



*3-[[3-[3,5-Bis(trifluoromethyl)phenyl]-1-(4-carboxyphenyl)pyrazol-4-yl]methylamino] benzoic acid* (**12**). Light gold solid; (458 mg, 83%); IR (KBr pellet, cm^−1^): 3361, 3064, 1685, 1604, 1520, 1176, 1132; ^1^H NMR, 300 MHz (DMSO-*d*_6_): δ 8.84 (s, 1H), 8.43 (s, 2H), 8.12–8.08 (m, 5H), 7.28–7.20 (m, 3H), 6.92 (s, 1H), 6.33 (br, 1H), 4.32 (s, 2H); ^13^C NMR (75 MHz, DMSO-d_6_): δ 168.3, 167.0, 148.9, 148.6, 142.4, 135.3, 131.8, 131.4 (^3^*J*_C-F_ = 4.5 Hz), 130.8 (^2^*J*_C-F_ = 32.7 Hz), 129.4, 129.0, 127.9, 123.6 (^1^*J*_C-F_ = 271.2 Hz), 122.0, 120.3, 118.4, 118.2, 118.0, 116.9, 113.5, 38.2. HRMS (ESI-FTMS Mass (*m*/*z*): calcd for C_26_H_17_F_6_N_3_O_4_ [M + H]^+^ = 550.1196, found 550.1194.



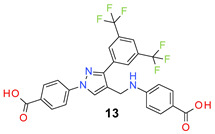



4-[[3-[3,5-Bis(trifluoromethyl)phenyl]-1-(4-carboxyphenyl)pyrazol-4-yl]methylamino] benzoic acid (**13**). Light yellow solid; (451 mg, 82%); IR (KBr pellet, cm^−1^): 3361, 2667, 1686,1601,1419, 1282, 1178; ^1^H NMR, 300 MHz (DMSO-*d*_6_): δ 8.84 (s, 1H), 8.39 (s, 2H), 8.13–8.08 (m, 5H), 7.71 (d, *J* = 7.1 Hz, 2H), 6.83 (s, 1H), 6.70 (d, *J* = 8.6 Hz, 2H), 4.36 (s, 2H); ^13^C NMR (75 MHz, DMSO-d_6_): δ 167.9, 167.0, 152.3, 148.9, 142.4, 135.3, 132.8, 131.5 (^3^*J*_C-F_ = 4.2 Hz), 130.8 (^2^*J*_C-F_ = 32.5 Hz), 129.1, 129.0, 127.9, 123.6 (^1^*J*_C-F_ = 271.2 Hz), 122.1, 119.9, 118.5, 118.2, 111.7, 37.7. HRMS (ESI-FTMS Mass (*m*/*z*): calcd for C_26_H_17_F_6_N_3_O_4_ [M + H]^+^ = 550.1196, found 550.1197.



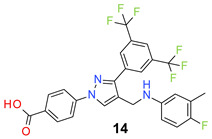



*4-[3-[3,5-Bis(trifluoromethyl)phenyl]-4-[(4-fluoro-3-methyl-anilino)methyl]pyrazol-1-yl] benzoic acid* (**14**). White solid; (497 mg, 92%); IR (KBr pellet, cm^−1^): 3424, 2906, 1698, 1608, 1358, 1281, 1177, 1139; ^1^H NMR, 300 MHz (DMSO-*d*_6_): δ 8.94 (s, 1H), 8.21 (s, 2H), 8.05 (t, *J* = 8.6 Hz, 5H), 6.94–6.83 (m, 3H), 4.48 (s, 2H), 2.04 (s, 3H); ^13^C NMR (75 MHz, DMSO-d_6_): δ 167.0, 158.3 (^1^*J*_C-F_ = 240.57 Hz), 150.0, 142.3, 134.9, 132.4, 131.5, 130.7 (^2^*J*_C-F_ = 32.7 Hz), 130.3, 129.4, 128.5, 125.6, 123.6 (^1^*J*_C-F_ = 271.11 Hz), 122.1, 118.6, 118.1, 115.9 (d, ^2^*J*_C-F_ = 23.5 Hz), 39.0, 14.5. HRMS (ESI-FTMS Mass (*m*/*z*): calcd for C_26_H_18_F_7_N_3_O_2_ [M + H]^+^ = 538.1360, found 538.1356.



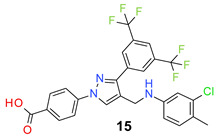



*4-[3-[3,5-Bis(trifluoromethyl)phenyl]-4-[(3-chloro-4-methyl-anilino)methyl]pyrazol-1-yl] benzoic acid* (**15**). Light brown solid; (453 mg, 81%); IR (KBr pellet, cm^−1^): 3366, 2930, 1724, 1608, 1405, 1277, 1177, 1128; ^1^H NMR, 300 MHz (DMSO-*d*_6_): δ 8.93 (s, 1H), 8.26 (s, 2H), 8.08–7.99 (m, 5H), 7.07 (d, *J* = 8.2 Hz, 1H), 6.95 (s, 1H), 6.80 (d, *J* = 8.0 Hz, 1H), 4.39 (s, 2H), 2.12 (s, 3H); ^13^C NMR (75 MHz, DMSO-d_6_): δ 166.9, 149.4, 142.3, 135.0, 133.8, 132.0, 131.8, 131.5, 130.8 (^2^*J*_C-F_ = 32.7 Hz), 129.2, 129.0, 128.2, 123.6 (^1^*J*_C-F_ = 271.1 Hz), 121.9, 118.5, 118.1, 117.5, 117.0, 116.6, 19.0. HRMS (ESI-FTMS Mass (*m*/*z*): calcd for C_26_H_18_F_6_N_3_O_2_Cl [M + H]^+^ = 554.1065, 556.1037, found 554.1065.



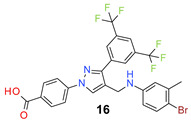



*4-[3-[3,5-Bis(trifluoromethyl)phenyl]-4-[(4-bromo-3-methyl-anilino)methyl]pyrazol-1-yl] benzoic acid* (**16**). Yellowish white solid; (489 mg, 82%); IR (KBr pellet, cm^−1^): 2930, 1683, 1607, 1355, 1277, 1134; ^1^H NMR, 300 MHz (DMSO-*d*_6_): δ 8.84 (s, 1H), 8.25 (s, 2H), 8.03–7.99 (m, 5H), 7.24 (d, *J* = 8.5 Hz, 1H), 6.80 (s, 1H), 6.61 (d, *J* = 7.9 Hz, 1H), 4.33 (s, 2H), 2.12 (s, 3H); ^13^C NMR (75 MHz, DMSO-d_6_): δ 167.0, 149.3, 143.8, 142.3, 138.0, 135.1, 132.8, 131.7, 131.4, 130.8 (^2^*J*_C-F_ = 32.7 Hz),129.2, 129.0, 128.1, 123.5 (^1^*J*_C-F_ = 271.2 Hz), 118.8, 118.4, 118.1, 117.7, 116.2, 22.9. HRMS (ESI-FTMS Mass (*m*/*z*): calcd for C_26_H_18_BrF_6_N_3_O_2_ [M + H]^+^ = 598.0559, 600.0540, found 598.0553.



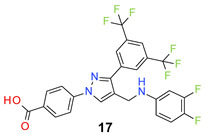



*4-[3-[3,5-Bis(trifluoromethyl)phenyl]-4-[(3,4-difluoroanilino)methyl]pyrazol-1-yl] benzoic acid* (**17**). White solid; (449 mg, 83%); IR (KBr pellet, cm^−1^): 3412, 2802, 1701, 1608, 1358, 1281, 1174, 1139; ^1^H NMR, 300 MHz (DMSO-*d*_6_): δ 8.83 (s, 1H), 8.35 (s, 2H), 8.08–8.05 (m, 5H), 7.19–7.09 (m, 1H), 6.74–6.71 (m, 1H), 6.51 (d, *J* = 6.6 Hz, 1H), 4.26 (s, 2H); ^13^C NMR (75 MHz, DMSO-d_6_): δ 167.0, 150.3 (dd, *J*_C-F_ = 13.1, 233.8 Hz), 149.0, 144.9 (d, ^3^*J*_C-F_ = 8.5 Hz), 142.5 (dd, *J*_C-F_ = 13.0, 239.0 Hz), 142.4, 135.2, 131.8, 131.4 (^3^*J*_C-F_ = 5.3 Hz), 130.8 (^2^*J*_C-F_ = 32.6 Hz), 129.1, 127.9, 123.6 (^1^*J*_C-F_ = 271.2 Hz), 123.2, 121.9, 119.3, 118.4, 117.9 (d, ^2^*J*_C-F_ = 17.4 Hz), 109.7, 102.0 (d, ^2^*J*_C-F_ = 20.2 Hz). HRMS (ESI-FTMS Mass (*m*/*z*): calcd for C_25_H_15_F_8_N_3_O_2_ [M + H]^+^ = 542.1109, found 542.1122.



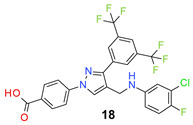



*4-[3-[3,5-Bis(trifluoromethyl)phenyl]-4-[(3-chloro-4-fluoro-anilino)methyl]pyrazol-1-yl] benzoic acid* (**18**). Off white solid (506 mg, 90%); IR (KBr pellet, cm^−1^): 3412, 2898, 1698, 1608, 1356, 1278, 1177, 1142. ^1^H NMR, 300 MHz (DMSO-*d*_6_): δ 8.85 (s, 1H), 8.34 (s, 2H), 8.09–8.05 (m, 5H), 7.14 (t, *J* = 9.0 Hz, 1H), 6.88 (s, 1H), 6.71 (d, *J* = 8.8 Hz, 1H), 4.29 (s, 2H); ^13^C NMR (75 MHz, DMSO-d_6_): δ 167.0, 150.8 (d, ^1^*J*_C-F_ = 235.0 Hz), 149.1, 144.3, 142.4, 135.2, 131.8, 131.4, 131.2 (^2^*J*_C-F_ = 32.7 Hz), 130.5, 129.1 (d, ^3^*J*_C-F_ = 10.5 Hz), 128.0, 123.6 (^1^*J*_C-F_ = 271.2 Hz), 122.0, 119.9 (d, ^2^*J*_C-F_ = 18.1 Hz), 119.0, 118.4, 117.4 (d, ^2^*J*_C-F_ = 21.4 Hz), 114.8. HRMS (ESI-FTMS Mass (*m*/*z*): calcd for C_25_H_15_ClF_7_N_3_O_2_ [M + H]^+^ = 558.0814, 560.0786, found 558.0816.



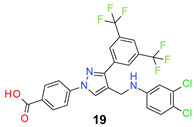



*4-[3-[3,5-Bis(trifluoromethyl)phenyl]-4-[(3,4-dichloroanilino)methyl]pyrazol-1-yl] benzoic acid* (**19**). Off white solid; (497 mg, 86%); IR (KBr pellet, cm^−1^): 3403, 2866, 1682, 1607, 1359, 1278, 1143; ^1^H NMR, 300 MHz (DMSO-*d*_6_): δ 8.83 (s, 1H), 8.39 (s, 2H), 8.14–8.08 (m, 5H), 7.29 (d, *J* = 8.7 Hz, 1H), 6.88–6.87 (m, 1H), 6.66 (dd, *J* = 2.5, 8.82 Hz, 1H), 6.54–6.52 (m, 1H), 4.31–2.29 (m, 2H); ^13^C NMR (75 MHz, DMSO-d_6_): δ 167.0, 152.3, 148.8, 148.3, 146.1, 142.4, 135.2, 131.4, 130.7 (^2^*J*_C-F_ = 35.3 Hz), 129.0, 127.8, 123.6 (^1^*J*_C-F_ = 266.3 Hz), 120.0 (^3^*J*_C-F_ = 5.9 Hz), 119.7, 118.3, 117.4, 117.1, 113.0, 112.6 (m), 38.4. HRMS (ESI-FTMS Mass (*m*/*z*): calcd for C_25_H_15_Cl_2_F_6_N_3_O_2_ [M + H]^+^ = 574.0518, 576.0490, found 574.0776.



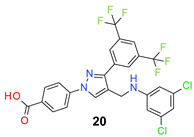



*4-[3-[3,5-Bis(trifluoromethyl)phenyl]-4-[(3,5-dichloroanilino)methyl]pyrazol-1-yl] benzoic acid* (**20**). Off white solid; (489 mg, 85%); IR (KBr pellet, cm^−1^): 3425, 3090, 1697, 1607, 1362, 1281, 1186, 1138; ^1^H NMR, 300 MHz (DMSO-*d*_6_): δ 8.82 (s, 1H), 8.36 (s, 2H), 8.12–8.07 (m, 5H), 6.67 (s, 4H), 4.32 (s, 2H); ^13^C NMR (75 MHz, DMSO-d_6_): δ 167.0, 150.7, 148.9, 142.4, 135.2, 134.8, 131.8, 131.4 (^3^*J*_C-F_ = 3.0 Hz),130.8 (^2^*J*_C-F_ = 32.8 Hz), 129.1, 127.9, 123.6 (^1^*J*_C-F_ = 271.2 Hz), 122.1, 119.8, 118.4, 115.4, 110.8, 37.8. HRMS (ESI-FTMS Mass (*m*/*z*): calcd for C_25_H_15_Cl_2_F_6_N_3_O_2_ [M + H]^+^ = 574.0518, 576.0490, found 574.0509.



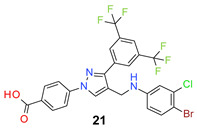



*4-[3-[3,5-Bis(trifluoromethyl)phenyl]-4-[(4-bromo-3-chloro-anilino)methyl]pyrazol-1-yl] benzoic acid* (**21**). Light yellow solid; (569 mg, 92%); IR (KBr pellet, cm^−1^): 3458, 3087, 1704, 1606, 1317, 1278, 1186, 1135; ^1^H NMR, 300 MHz (DMSO-*d*_6_): δ 8.81 (s, 1H), 8.35 (s, 2H), 8.09–8.05 (m, 5H), 7.37 (d, *J* = 5.7 Hz, 1H), 6.86 (s, 1H), 6.58 (d, *J* = 8.4 Hz, 1H), 4.25 (s, 2H); ^13^C NMR (75 MHz, DMSO-d_6_): δ 167.0, 149.2, 148.8, 142.4, 135.2, 134.0, 133.6, 131.8, 131.4 (^3^*J*_C-F_ = 3.1 Hz), 130.8 (^2^*J*_C-F_ = 32.7 Hz), 129.1, 127.9, 123.6 (^1^*J*_C-F_ = 271.2 Hz), 122.0, 119.9, 118.4, 113.7, 113.5, 106.5, 38.0. HRMS (ESI-FTMS Mass (*m*/*z*): calcd for C_25_H_15_BrClF_6_N_3_O_2_ [M + H]^+^ = 618.0013, 619.9993, found 619.9990.



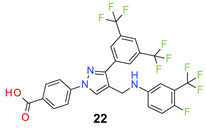



*4-[3-[3,5-Bis(trifluoromethyl)phenyl]-4-[[4-fluoro-3-(trifluoromethyl)anilino]methyl]pyrazol-1-yl] benzoic acid* (**22**). White solid; (508 mg, 86%); IR (KBr pellet, cm^−1^): 3429, 3093, 1701, 1608, 1422, 1282, 1154; ^1^H NMR, 300 MHz (DMSO-*d*_6_): δ 8.83 (s, 1H), 8.37 (s, 2H), 8.06 (s, 5H), 7.22 (br, 1H), 6.93 (s, 2H), 6.44 (s, 1H), 4.30 (s, 2H); ^13^C NMR (75 MHz, DMSO-d_6_): δ 167.0, 152.6 (d, ^1^*J*_C-F_ = 249.0 Hz), 148.9, 145.4, 142.45, 135.2, 131.8, 131.4 (^3^*J*_C-F_ = 3.8 Hz), 130.8 (^2^*J*_C-F_ = 32.7 Hz), 128.9 (^2^*J*_C-F_ = 32.4 Hz), 127.9, 123.6 (^1^*J*_C-F_ = 271.1 Hz), 122.0 (^3^*J*_C-F_ = 3.6 Hz), 123.3 (^1^*J*_C-F_ = 270.1 Hz), 120.0, 118.4, 117.9 (d, ^2^*J*_C-F_ = 21.1 Hz), 117.5 (d, ^3^*J*_C-F_ = 7.2 Hz), 117.0 (dd, *J*_C-F_ = 13.5, 24.0 Hz), 109.4 (d, ^4^*J*_C-F_ = 4.6 Hz), 38.3. HRMS (ESI-FTMS Mass (*m*/*z*): calcd for C_26_H_15_F_10_N_3_O_2_ [M + H]^+^ = 592.1077, found 592.1082.



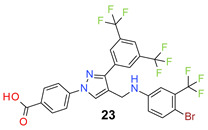



*4-[3-[3,5-Bis(trifluoromethyl)phenyl]-4-[[4-bromo-3-(trifluoromethyl)anilino]methyl]pyrazol-1-yl] benzoic acid* (**23**). White solid; (543 mg, 83%); IR (KBr pellet, cm^−1^): 3454, 2983, 1682, 1605, 1439, 1278, 1142; ^1^H NMR, 300 MHz (DMSO-*d*_6_): δ 8.83 (s, 1H), 8.36 (s, 2H), 8.11–8.07 (m, 5H), 7.51 (d, *J* = 8.7 Hz, 1H), 7.08 (s, 1H), 6.93–6.81 (m, 2H), 4.35 (s, 2H); ^13^C NMR (75 MHz, DMSO-d_6_): δ 167.0, 148.9, 148.1, 142.4, 135.5 (^3^*J*_C-F_ = 29.2 Hz), 131.4 (^3^*J*_C-F_ = 4.8 Hz), 130.8 (^2^*J*_C-F_ = 32.4 Hz), 129.1, 128.8, 127.9 (^3^*J*_C-F_ = 6.1 Hz), 123.6 (^1^*J*_C-F_ = 271.1 Hz), 122.1, 120.1, 119.8, 118.5, 117.0, 116.2, 111.9, 103.2, 37.9. HRMS (ESI-FTMS Mass (*m*/*z*): calcd for C_26_H_15_BrF_9_N_3_O_2_ [M + H]^+^ = 652.0277, 654.0258, found 652.0280.



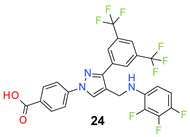



*4-[3-[3,5-Bis(trifluoromethyl)phenyl]-4-[(2,3,4-trifluoroanilino)methyl]pyrazol-1-yl] benzoic acid* (**24**). White solid; (433 mg, 77%); IR (KBr pellet, cm^−1^): 3413, 3090, 1694, 1608, 1359, 1281, 1174, 1131; ^1^H NMR, 300 MHz (DMSO-*d*_6_): δ 8.74 (s, 1H), 8.39 (s, 2H), 8.12–8.06 (m, 5H), 7.07–7.04 (m, 1H), 6.60–6.58 (m, 1H), 6.17 (s, 1H), 4.39 (s, 2H); ^13^C NMR (75 MHz, DMSO-d_6_): δ 167.0, 148.8, 142.5, 142.3 (dd, *J*_C-F_ = 10.9, 239.0 Hz), 140.1 (dt, *J*_C-F_ = 7.2, 234.3 Hz), 139.6 (d, ^1^*J*_C-F_ = 242.7 Hz), 135.5, 134.7 (d, ^3^*J*_C-F_ = 9.6 Hz), 131.8, 131.4 (^3^*J*_C-F_ = 5.0 Hz), 130.6 (^2^*J*_C-F_ = 32.6 Hz), 129.0, 128.1 (d, ^4^*J*_C-F_ = 2.7 Hz), 123.6 (^1^*J*_C-F_ = 271.1 Hz), 121.9, 120.3, 118.4, 111.8 (dd, *J*_C-F_ = 3.2, 18.7 Hz), 106.6, 38.3. HRMS (ESI-FTMS Mass (*m*/*z*): calcd for C_25_H_14_F_9_N_3_O_2_ [M + H]^+^ = 560.1015, found 560.1030.



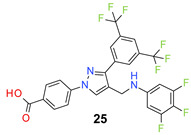



*4-[3-[3,5-Bis(trifluoromethyl)phenyl]-4-[(3,4,5-trifluoroanilino)methyl]pyrazol-1-yl] benzoic acid* (**25**). White solid; (427 mg, 76%); IR (KBr pellet, cm^−1^): 3420, 1689, 1608, 1359, 1279, 1178, 1135; ^1^H NMR, 300 MHz (DMSO-d_6_): δ 8.82 (s, 1H), 8.37 (s, 2H), 8.12–8.08 (m, 5H), 6.54 (s, 3H), 4.29 (s, 2H); ^13^C NMR (75 MHz, DMSO-*d*_6_): δ 167.0, 152.0 (dt, *J* = 10.2, 233.5 Hz), 148.9, 145.5 (t, *J* = 12.1 Hz), 142.4, 135.2, 132.1 (^2^*J*_C-F_ = 38.8 Hz), 131.5, 130.9 (^3^*J*_C-F_ = 8.5 Hz), 130.5, 129.2, 127.9, 123.6 (^1^*J*_C-F_ = 271.1 Hz), 122.1, 119.7, 118.5, 96.1 (d, ^2^*J*_C-F_ = 22.9 Hz), 38.1. HRMS (ESI-FTMS Mass (*m*/*z*): calcd for C_25_H_14_F_9_N_3_O_2_ [M + H]^+^ = 560.1015, found 560.1026.



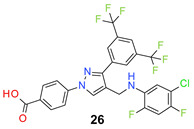



*4-[3-[3,5-Bis(trifluoromethyl)phenyl]-4-[(5-chloro-2,4-difluoro-anilino)methyl] pyraz-ol-1-yl] benzoic acid* (**26**). Light yellow solid; (477 mg, 83%); IR (KBr pellet, cm^−1^): 3446, 3061, 1682, 1608, 1364, 1277, 1175, 1132; ^1^H NMR, 300 MHz (DMSO-*d*_6_): δ 8.74 (s, 1H), 8.39 (s, 2H), 8.12–8.06 (m, 5H), 7.37 (t, *J* = 9.2 Hz, 1H), 6.96 (t, *J* = 8.3 Hz, 1H), 6.11 (s, 1H), 4.38 (d, *J* = 4.7 Hz, 2H); ^13^C NMR (75 MHz, DMSO-d_6_): δ 167.0, 149.3 (dd, *J*_C-F_ = 10.3, 246.6 Hz), 148.9, 148.3 (dd, *J*_C-F_ = 11.3, 240.7 Hz), 142.5, 135.4, 134.5 (d, ^3^*J*_C-F_ = 10.7 Hz), 131.8, 131.4 (^3^*J*_C-F_ = 6.7 Hz), 130.7 (^2^*J*_C-F_ = 32.7 Hz), 129.0, 128.1, 123.6 (^1^*J*_C-F_ = 271.1 Hz), 120.2, 118.4, 115.0 (d, ^2^*J*_C-F_ = 21.6 Hz), 112.6 (m), 105.2 (t, *J*_C-F_ = 25.0 Hz), 79.4 (t, *J*_C-F_ = 32.8 Hz), 38.2. HRMS (ESI-FTMS Mass (*m*/*z*): calcd for C_25_H_14_ClF_8_N_3_O_2_ [M + H]^+^ = 576.0720, 578.0692, found 576.0721.



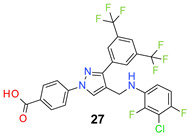



*4-[3-[3,5-Bis(trifluoromethyl)phenyl]-4-[(3-chloro-2,4-difluoro-anilino)methyl]pyraz-ol-1-yl] benzoic acid* (**27**). Off white solid; (399 mg, 69%); IR (KBr pellet, cm^−1^): 3425, 2875, 1694, 1608, 1359, 1278, 1173, 1128; ^1^H NMR, 300 MHz (DMSO-*d*_6_): δ 8.74 (s, 1H), 8.39 (s, 2H), 8.12–8.06 (m, 5H), 7.07 (t, *J* = 8.9 Hz, 1H), 6.83–6.78 (m, 1H), 6.11 (s, 1H), 4.41 (s, 2H); ^13^C NMR (75 MHz, DMSO-d_6_): δ 167.0, 149.6 (d, ^1^*J*_C-F_ = 235.9 Hz), 148.8, 147.6 (d, ^1^*J*_C-F_ = 244.6 Hz), 142.5, 135.5, 134.3 (d, ^3^*J*_C–F_ = 11.1 Hz), 131.8, 131.4 (^3^*J*_C-F_ = 7.7 Hz), 130.7 (^2^*J*_C-F_ = 25.6 Hz), 129.0, 128.1, 123.6 (^1^*J*_C-F_ = 271.1 Hz), 122.0, 120.4, 118.4, 111.7 (d, ^2^*J*_C-F_ = 23.1 Hz), 111.0 (d, ^4^*J*_C-F_ = 4.6 Hz), 108.5 (d, ^2^*J*_C-F_ = 21.5 Hz), 38.3. HRMS (ESI-FTMS Mass (*m*/*z*): calcd for C_25_H_14_ClF_8_N_3_O_2_ [M + H]^+^ = 576.0720, 578.0692, found 576.0729.



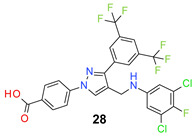



*4-[3-[3,5-Bis(trifluoromethyl)phenyl]-4-[(3,5-dichloro-4-fluoro-anilino)methyl]pyraz-ol-1-yl] benzoic acid* (**28**). Light yellow solid; (533 mg, 90%); IR (KBr pellet, cm^−1^): 3409, 3087, 1686, 1607, 1359, 1282, 1187, 1143; ^1^H NMR, 300 MHz (DMSO-*d*_6_): δ 8.81 (s, 1H), 8.36 (s, 2H), 8.12–8.07 (m, 5H), 6.79 (d, *J* = 5.6 Hz, 2H), 6.49 (s, 1H), 4.30 (d, *J* = 3.8 Hz, 2H); ^13^C NMR (75 MHz, DMSO-d_6_): δ 167.0, 148.8, 146.0, 145.3 (d, ^1^*J*_C-F_ = 233.6 Hz), 143.8, 135.2, 131.8, 131.4 (^3^*J*_C-F_ = 3.2 Hz), 130.8 (^2^*J*_C-F_ = 32.7 Hz), 129.1 (d, ^3^*J*_C-F_ = 7.1 Hz), 127.9, 123.6 (^1^*J*_C-F_ = 271.2 Hz), 122.0, 121.5 (d, ^2^*J*_C-F_ = 17.7 Hz), 119.8, 118.4, 118.2, 112.1, 79.4 (t, *J*_C-F_ = 33.0 Hz), 38.1. HRMS (ESI-FTMS Mass (*m*/*z*): calcd for C_25_H_14_Cl_2_F_7_N_3_O_2_ [M + H]^+^ = 592.0424, 594.0396, found 592.0402.



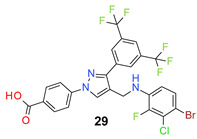



*4-[3-[3,5-Bis(trifluoromethyl)phenyl]-4-[(4-bromo-3-chloro-2-fluoro-anilino)methyl]pyrazol-1-yl] benzoic acid* (**29**). Light yellow solid; (558 mg, 88%); IR (KBr pellet, cm^−1^): 3464, 2860, 1686, 1601, 1424, 1064; ^1^H NMR, 300 MHz (DMSO-*d*_6_): δ 8.72 (s, 1H), 8.37 (s, 2H), 8.14–8.07 (m, 5H), 7.32 (d, *J* = 8.8 Hz, 1H), 6.75 (t, *J* = 8.5 Hz, 1H), 6.49 (s, 1H), 4.44 (d, *J* = 4.5 Hz, 2H); ^13^C NMR (75 MHz, DMSO-d_6_): δ 167.0, 148.7, 147.4 (d, ^1^*J*_C-F_ = 243.0 Hz), 142.5, 137.2 (d, ^3^*J*_C-F_ = 12.0 Hz), 135.5, 131.4, 130.6 (^2^*J*_C-F_ = 33.2 Hz), 129.0, 128.7 (^3^*J*_C-F_ = 3.0 Hz), 128.1, 123.6 (^1^*J*_C-F_ = 271.1 Hz), 122.0, 120.4 (d, ^2^*J*_C-F_ = 24.6 Hz), 118.4, 117.1, 112.6, 106.8, 103.2 (t, *J*_C-F_ = 20.9 Hz), 38.1. HRMS (ESI-FTMS Mass (*m*/*z*): calcd for C_25_H_14_BrClF_7_N_3_O_2_ [M + H]^+^ = 635.9919, 637.9899, found 637.9897.



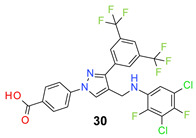



*4-[3-[3,5-Bis(trifluoromethyl)phenyl]-4-[(3,5-dichloro-2,4-difluoro-anilino)methyl]pyrazol-1-yl] benzoic acid* (**30**). Off white solid; (413 mg, 67%); IR (KBr pellet, cm^−1^): 3427, 3088, 1694, 1608, 1360, 1278, 1178, 1135; ^1^H NMR, 300 MHz (DMSO-*d*_6_): δ 8.72 (s, 1H), 8.36 (s, 2H), 8.12–8.06 (m, 5H), 6.99 (t, *J* = 7.6 Hz, 1H), 6.45 (d, *J* = 4.32 Hz, 1H), 4.45 (s, 2H); ^13^C NMR (75 MHz, DMSO-d_6_): δ 167.0, 148.8, 145.8 (d, ^1^*J*_C-F_ = 243.7 Hz), 144.9 (d, ^1^*J*_C-F_ = 235.9 Hz), 142.5, 135.5, 134.6 (dd, *J*_C-F_ = 7.5, 10.0 Hz), 131.8, 131.4 (d, ^3^*J*_C-F_ = 6.1 Hz), 130.6 (^2^*J*_C-F_ = 36.8 Hz), 129.0, 128.1, 123.6 (^1^*J*_C-F_ = 271.4 Hz), 122.0, 120.2, 118.5, 116.2 (d, ^2^*J*_C-F_ = 19.1 Hz), 110.7 (^3^*J*_C-F_ = 3.7 Hz), 109.8 (t, *J*_C-F_ = 19.7 Hz), 38.0. HRMS (ESI-FTMS Mass (*m*/*z*): calcd for C_25_H_13_Cl_2_F_8_N_3_O_2_ [M + H]^+^ = 610.0330, 612.0302, found 610.0322.

### 3.5. Minimum Inhibitory Concentration Assay (MIC)

As previously reported, the MIC values of the substances were calculated using the standard microdilution technique suggested by the Clinical and Laboratory Standards Institute (CLSI) [36]. Compounds were dissolved in dimethyl sulfoxide (DMSO) to achieve a starting concentration of 32 µg/mL as described by us previously [29,31]. Compounds were further diluted two-fold across the 96-well honeycomb plate columns in cation adjusted Mueller Hinton broth (CAMHB), and bacterial culture was added. Plates were incubated for 18–20 h at 35 ℃ before determining the minimum concentrations at which wells had no visible turbidity to the naked eye. The MIC values were determined from at least a duplicate result obtained from three independent experiments using fresh bacterial cultures on different days.

### 3.6. Cytotoxicity Assay

The human embryonic kidney cell line (HEK-293) was used to assess the cytotoxicity of the antibiotics as described by us previously [29,31]. HEK-293 cells were cultured in Dulbecco’s modified eagle’s medium (DMEM) containing 10% Fetal Bovine Serum (FBS) and incubated at 37 °C in the presence of 5% carbon dioxide. HEK-293 cells were cultured in 96-well black plates with 6000 cells per well for the first 24 h followed by treatment with a range of concentrations of antibiotics dissolved in DMSO for the next 24 h. After the second period of incubation, resazurin (40 µL of 0.15 mg/mL) was added to each well and incubated for an additional 4 h. Reduction of resazurin was measured fluorometrically with excitation wavelength at 560 nm and emission at 590 nm using BioTek^™^ Cytation^™^5 plate reader. Percentage viability of HEK cells for each range of concentrations was calculated, taking the fluorescence value of the wells treated with DMSO as a reference, using Microsoft^®^ Excel^®^ for Office 365 MSO. IC_50_ values were computed using the calculated percentage viability values through data processing in Graphpad Prism 7 for Windows, Version 7.04.

### 3.7. Time Kill Assay

Time Kill Assay was performed against *S. aureus* ATCC 33599 using our potent compounds following methodology reported previously with few modifications [29,31]. The bacterial strain was grown to its logarithmic growth phase by shaking at 200 rpm in a shaking incubator at 35 °C. The culture was diluted to ~1.5 × 10^7^ CFU/mL in CAMHB and treated with concentrations equivalent to 4× MIC of tested compounds, including vancomycin and daptomycin, as control drugs. Aliquots were collected from treatment every two hours of incubation at 35 °C up to 12 h and one in 24 h, then serially diluted in sterile 1× PBS. The diluted aliquots were then transferred to blood agar plates by using the 6 × 6 drop plate method and incubated at 35 °C for 18–20 h prior to colony counting to obtain CFU/mL. Time-kill line graphs (CFU/mL vs. time) were constructed using Microsoft^®^ Excel^®^ for Office 365 MSO.

### 3.8. Persister Assay

The persister assay was performed following the methodology described previously [29]. Briefly, for persister viability assays, *S. aureus* ATCC 700699 strain (MRSA) was grown to stationary phase in CAMHB medium by shaking at 200 rpm in a shaking incubator for 24 h at 35 °C. An aliquot of stationary phase culture was washed and diluted to ~5 × 10^8^ CFU/mL in 1×PBS then treated with 32× MIC of each compound and control drugs in sterile 10 × 75 mm plastic culture tubes and incubated at 35 °C with shaking at 200 rpm for 4 h. After incubation, 400 µL aliquot was washed, serially diluted in PBS, and plated in blood agar plate for viable colony count.

Likewise, for persister time-kill assays, the persister cells were treated at varied concentrations and incubated with shaking. At the start and every 2 h until 8 h, 400 µL samples were washed, serially diluted, and plated on blood agar using the 6 × 6 drop plate method. Plates were incubated for 18–20 h at 35 °C, before the colonies were counted to calculate CFU/mL. All the treatments were performed in triplicates.

### 3.9. Calgary Biofilm Device

Minimum biofilm eradication concentration (MBEC) was determined using a Calgary device using the protocol reported elsewhere with some modifications [37]. The Calgary device is a 96-well plate with a lid containing pegs to establish biofilms. Bacteria from an overnight plate culture were suspended to a 0.5 McFarland turbidity, then diluted 1:100 in a CAMHB containing 1% glucose which was inoculated in the 96 wells (150 μL in each well) and incubated at 35 °C for 24 h to establish biofilm on the pegs. The lid was then removed, washed with PBS in a fresh 96 well plate, then transferred to another 96-well plate containing 2-fold serial dilutions of the test compounds dissolved in PBS (150 μL total volume in each well). This plate was called the “challenge plate”. The challenge plate was incubated for an additional 24 h, and the lid was transferred to the next 96-well plate containing fresh CAMHB media with 1% glucose (180 μL) and further incubated for 24 h. After this final incubation, the wells were observed for visible turbidity. MBEC was the well with the lowest concentration of compound that resulted in no turbidity. All the compounds were tested in a minimum of three independent experiments.

## 4. Conclusions

In summary, we are reporting an efficient synthesis of 30 novel pyrazole derivatives. Most of these compounds are potent growth inhibitors of Gram-positive bacterial strains with MIC values as low as 0.25 µg/mL. These bactericidal compounds are potent against MRSA persisters. Three of the four compounds tested for their biofilm eradication property showed potent activity. Enthusiasm was a little bit dampened, due to the toxicity of the compounds for human cell lines. Nevertheless, these potent compounds can be modified to increase their pharmacological properties and reduce their toxicity profile.

## Figures and Tables

**Figure 1 molecules-26-05083-f001:**
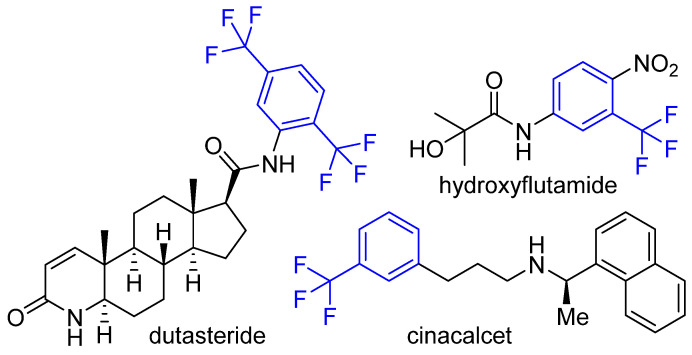
Representative examples of trifluoromethyl phenyl-containing approved drugs.

**Figure 2 molecules-26-05083-f002:**
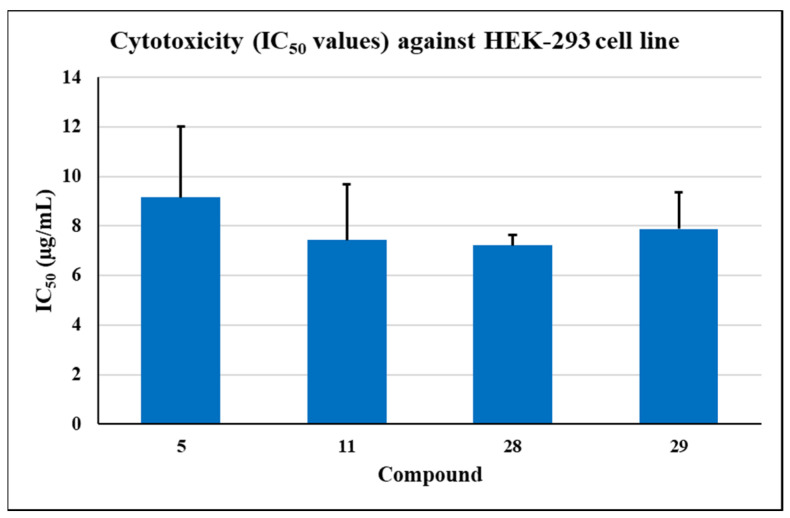
Cytotoxicity of potent antimicrobials for human embryonic kidney (HEK-293) cells. The data show the average of the IC_50_ values obtained from triplicate assays for each compound. Error bars indicate the standard deviation values (*n* = 3).

**Figure 3 molecules-26-05083-f003:**
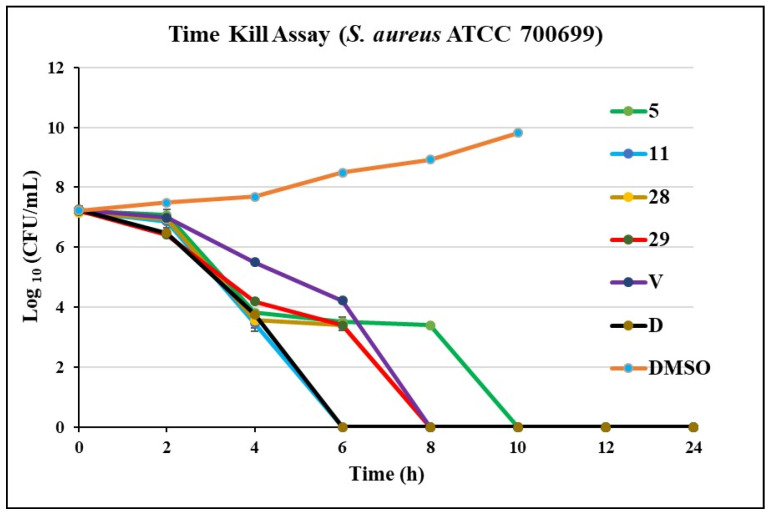
Time Kill Assay of potent compounds **5**, **11**, **28**, and **29** against an MRSA (*S. aureus* ATCC 700699), Vancomycin (**V**), and Daptomycin (**D**) are positive controls. All compounds, including positive controls, were tested at a 4× MIC concentration.

**Figure 4 molecules-26-05083-f004:**
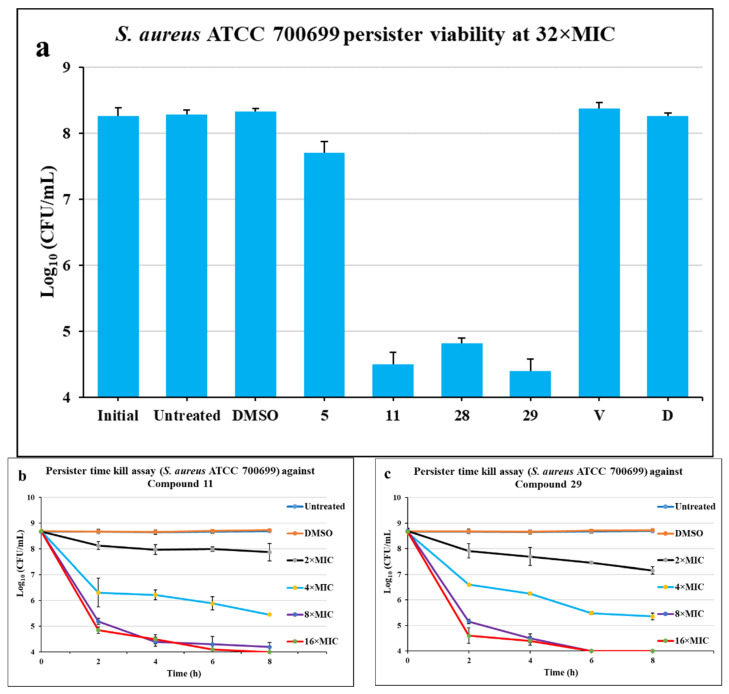
The activity of potent compounds against MRSA persisters. (**a**) Persister viability assay at 32× MIC for each compound run for 4 h. Persister Time Kill Assay against (**b**) compound **11** and (**c**) compound **29** at various MIC values run for 8 h. The error bars represent the standard deviation of the colony count performed in triplicates for each treatment, *n* = 3.

**Table 1 molecules-26-05083-t001:** Synthesis of 3,5-bis(trifluoromethylphenyl)-derived pyrazole anilines.

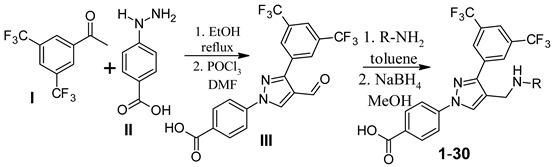
**#**	**R**	**HRMS** **(M + H)^+^**	**Yield (%)**	**#**	**R**	**HRMS** **(M + H)^+^**	**Yield (%)**
**1**	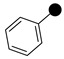	506.1291	86	**16**	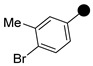	598.0553	82
**2**	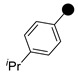	548.1761	80	**17**	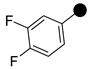	542.1122	83
**3**	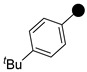	562.1919	93	**18**	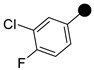	558.0816	90
**4**	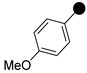	536.1397	77	**19**	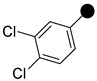	560.0776	86
**5**	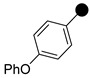	598.1556	79	**20**	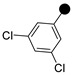	574.0509	85
**6**	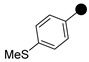	552.1173	70	**21**	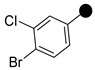	619.9990	92
**7**	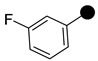	524.1208	82	**22**	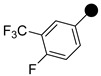	592.1082	86
**8**	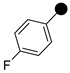	524.1200	84	**23**	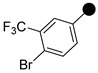	652.0280	83
**9**	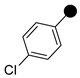	540.0907	72	**24**	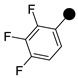	560.1030	77
**10**	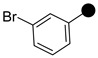	584.0410	85	**25**	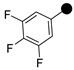	560.1026	76
**11**	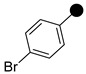	584.0408	78	**26**	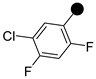	576.0721	83
**12**	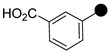	550.1194	83	**27**	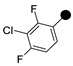	576.0729	69
**13**	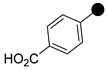	550.1197	82	**28**	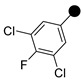	592.0402	90
**14**	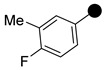	538.1356	92	**29**	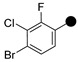	637.9897	88
**15**	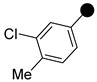	554.1065	81	**30**	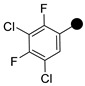	610.0322	67

**Table 2 molecules-26-05083-t002:** MIC (µg/mL) data of the synthesized compounds. Gram-positive: Antibiotic susceptible *S. aureus* ATCC 25923 (Sa23), antibiotic-resistant (methicillin-resistant) strains: *S. aureus* BAA-2312 (Sa12), *S. aureus* ATCC 33591 (Sa91), *S. aureus* ATCC 33592 (Sa92), and *S. aureus* ATCC 700699 (Sa99); *S. epidermidis* ATCC 700296 (Se), antibiotic susceptible *E. faecalis* ATCC 29212 (Ef12), antibiotic-resistant (vancomycin-resistant, VRE) *E. faecalis* ATCC 51299 (Ef99), antibiotic-resistant (vancomycin-resistant, VRE) *E. faecium* ATCC 700221 (Ef21), *Bacillus subtilis* ATCC 6623 (Bs), V = Vancomycin and D = Daptomycin, *n* = 3.

#	Sa23	Sa12	Sa91	Sa92	Sa99	Se	Ef12	Ef99	Ef21	Bs
**1**	4	2	4	4	4	8	8	4	4	2
**2**	2	1	1	1	2	2	2	1	1	1
**3**	1	1	1	2	2	1	1	0.5	2	2
**4**	8	4	8	8	8	16	16	16	8	4
**5**	1	2	1	1	2	4	2	1	1	2
**6**	2	2	2	2	4	4	4	2	2	1
**7**	4	2	4	2	4	4	8	4	4	2
**8**	2	2	2	2	4	4	4	4	4	1
**9**	2	1	2	1	2	2	2	2	2	1
**10**	2	2	2	2	2	4	4	2	4	2
**11**	2	1	1	1	2	2	2	1	1	1
**12**	>32	>32	>32	>32	>32	>32	>32	>32	>32	>32
**13**	>32	32	>32	>32	>32	>32	>32	>32	>32	>32
**14**	2	1	2	1	2	4	2	2	1	1
**15**	2	1	1	2	2	2	2	2	1	1
**16**	1	1	1	1	2	2	1	2	1	1
**17**	2	2	2	2	2	4	4	4	2	1
**18**	2	1	1	1	2	2	2	2	1	1
**19**	1	0.5	0.5	1	2	1	1	1	1	1
**20**	1	1	0.5	1	1	2	1	1	1	1
**21**	1	1	0.5	0.5	2	2	1	1	1	1
**22**	1	1	1	1	1	2	2	2	1	2
**23**	1	2	1	1	2	4	2	2	2	1
**24**	2	1	1	1	2	4	2	2	1	1
**25**	4	2	2	2	4	4	4	4	2	1
**26**	2	0.25	2	1	1	4	1	2	0.25	0.25
**27**	1	1	1	1	2	2	2	2	1	1
**28**	1	1	1	1	1	2	2	2	1	1
**29**	2	1	1	1	2	2	2	2	1	2
**30**	1	2	1	1	2	2	2	2	1	1
**V**	1	0.5	2	2	4	2	2	>32	>32	0.25
**D**	2	2	2	2	8	2	16	16	16	2

**Table 3 molecules-26-05083-t003:** Biofilm studies of potent compounds.

#	MBEC (µg/mL)	
	Sa23	Efs12
**5**	>32	>32
**11**	2	4
**28**	1	2
**29**	2	2
**V**	>32	>32
**D**	>32	>32

## Data Availability

Not applicable.

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
