# Peer review of "Synthesis of 3,5-Bis(trifluoromethyl)phenyl-Substituted Pyrazole Derivatives as Potent Growth Inhibitors of Drug-Resistant Bacteria"

_molecules, 2021, doi:10.3390/molecules26165083_

Round 1

Reviewer 1 Report

The presented manuscript edited “Synthesis of 3,5-Bis(trifluoromethyl)phenyl-Substituted Pyra-2 zole Derivatives as Potent Growth Inhibitors of Drug-Resistant 3 Bacteria” deals with a current and essential topic. Drug-resistant ESKAPE is a significant problem in current medicines.

Overall, the manuscript was well organized with well-described.

However, some points need to be clarified before publication. Additionally, some analyses can significantly improve the manuscript, and it is worth considering their inclusion by the authors.

There are few suggestions to improve the manuscript:

General comments:

  1. In my opinion, the materials and method should be at the end of the manuscript in MDPI format.
  2. Determination of lipophilicity can provide much exciting information. The physicochemical properties of drug candidates should be the measurement in the early step of the drug discovery process since they allow to prediction of pharmacokinetic properties.

To summarize, the work may be published in molecules after revision.

Author Response

General comments:

  1. In my opinion, the materials and method should be at the end of the manuscript in MDPI format.

We moved the materials and method section at the end of the manuscript.

2. Determination of lipophilicity can provide much exciting information. The physicochemical properties of drug candidates should be the measurement in the early step of the drug discovery process since they allow to prediction of pharmacokinetic properties.

We are working on the physicochemical properties along with further pharmacological studies. Which will be reported soon.

Reviewer 2 Report

The manuscript „Synthesis of 3,5-Bis(trifluoromethyl)phenyl-Substituted Pyrazole Derivatives as Potent Growth Inhibitors of Drug-Resistant Bacteria “ by Ibrahim S. Alkhaibari and coworkers reports on the chemical synthesis of a series covering 3,5-bis(trifluoromethyl)phenyl-substituted pyrazole derivatives and their biological investigations to serve as antibacterial drug candidates.

@ line 24: why are some of the mentioned strains underlined?

@ line 35: “Alcohol tolerance” please clarify. The authors probably mean resistance against isopropanol used as disinfectant?

@ line 46: Probably it is better termed as 1H-Pyrazole?

@ “A number drugs containing pyrazole nucleus may be due to its decreased susceptibility to oxidative degradation metabolism compared to other five-membered heterocycles [14].” Please revise the grammer of this sentence.

@ line 57: replace “–CF3” by “trifluoromethyl (-CF3)”

@ line 62: reference 62 is not appropriate as this paper does not deal with TRifluoromethyl substituted phenyls

@ section 2: This is like “Materials and Methods” and should appear later in the manuscript. Please have a close look at the “Research Manuscript Sections” (https://www.mdpi.com/journal/molecules/instructions#preparation)

@ spectroscopic data: Please provide assignment of the most obvious IR bands and assignment of the protons of the 1H NMR spectrum for all compounds.

@ line 71: better say: 1H and 13C NMR spectroscopy, @ line 72: …NMR spectra

@ line 100: ??

The paragraph lines 101-107 should appear straight after describing the procedure (after line 89)

@ line 102/109/117/126/…: Bis (etc.) should be with capital letter.

@ “technique suggested by the Clinical and Laboratory Standards Institute (CLSI)” References 29 and 30 do not refer to CLSI but are paper of the research group. Please exchange the citation for appropriate ones.

@ line 444: “MBEC was the well…” It was rather the concentration in the well.

@ line 451: There are also redundant self-citations which do not describe the synthesis itself but its application.

@ line 455: “comprise” is a strange vocabulary: influence, impact, …

@ line 460: “…but no activity against the Gram-negative strains” Please provide reasons for the missing activity of the compounds against Gram-negative bacteria. Maybe this can be related to the mechanism of antibacterial action. The author do not provide any insight into the mode of action. This is important to overcome resistance and design prper drug candidates. The authors should provide studies on the mode of action in the manuscript.

@ Table 1: The size of the phenyl residues is different. Please unify.

@ Table 2: “Antimicrobial studies novel compounds” please revise this sentence. Explain the abbreviations V and D, and what the data mean, probably MIC values in µg/ml as determined by…assay. Please add these information to the caption. How many repeats were performed, n=? #6 appears two time in the table, #7 is missing: is this a typo?

@ line 497 and in the Supplementary Material: The use of “&” is not common in scientific papers; also @ line 536: “To our delight” is not scientific.

@ line 498 “several compounds”: please specify.

@ line 503: The authors discuss lipophilicity. Please provide data, e.g. logP value of the compounds.

@ Figure 2: n=?

@ Figure 3: It is hard to read due to usage of the same colors (green, orange) for two different compounds (although the color of the dots differ). Why was DMSO assessed after 12 and 24 h?

@ line 534: a number of efforts are being made ?

@ line 536: the four potent: Name the compounds using their numbers.

@ Figure 4: It is not necessary to insert differently colored dots to the prahs. This just confuses the reader. Use one color per compound.

@ line 562: This was already mentioned in line 34.

@ line 568: the abbreviation was already explained in line 432.

@ line 581: “Nevertheless, these potent compounds can be modified to increase their pharmacological properties and reduce their toxicity profile” This is an empty phrase as it is true for any drug.

The interpretation of the biological data (MIC) with respect to the chemical structure of the derivatives and the adducing of SAR is confusing. Please clarify this section and provide clear statements which chemical modifications are valuable for further drug optimization/design. What is the difference between (a) the halogen substituents, and (b) their different location at the phenyl residue?

The titel of the manuscript states “drug resistant bacteria”. I am wondering, since the authors did not only investigate resistant strains. Moreover, there is no information about the particular resistance of the used strains. Please provide information in the manuscript.

Although there will be a close editing by the publisher, the authors are kindly ask to correct some formal errors and inconsistency, such as (the following list is to be understood exemplarily): different style of presenting the authors’ names (@ line 5), typos (@ line 10), explanation of abbreviation (NMR is not explained, whereas FTIR is, please be consistent), @ line 81: 30 mL of…, @ line 91: 0.55 mmol of aniline derivative, remove abundant spaces in the nomenclature names (@ lines 102, 109) in the numeration (@ line 145, 146), use of space e.g., @ line 110, 198, 253, 254, 273, 274,… 378,…; different fonts e.g., @ line 295, 305, 314, 324,…, 366-373 ; @ line 389 vs. 396, 397 usage of capital letters?, @403: HEK-293; @ line 408 capital letters Time Kill Assay?, @ line 414: 2 h; @ line 417: using dash/hyphen time-kill and before without, also 96-well, e.g. lines 434/437/438/439;

Author Response

The manuscript „Synthesis of 3,5-Bis(trifluoromethyl)phenyl-Substituted Pyrazole Derivatives as Potent Growth Inhibitors of Drug-Resistant Bacteria “ by Ibrahim S. Alkhaibari and coworkers reports on the chemical synthesis of a series covering 3,5-bis(trifluoromethyl)phenyl-substituted pyrazole derivatives and their biological investigations to serve as antibacterial drug candidates.

@ line 24: why are some of the mentioned strains underlined?

The manuscript is focused on underlined strains.

@ line 35: “Alcohol tolerance” please clarify. The authors probably mean resistance against isopropanol used as disinfectant?

We added '(ethanol and isopropanol).

@ line 46: Probably it is better termed as 1H-Pyrazole?

We added the same.

@ “A number drugs containing pyrazole nucleus may be due to its decreased susceptibility to oxidative degradation metabolism compared to other five-membered heterocycles [14].” Please revise the grammer of this sentence.

Thank you for pointing out this. We corrected the sentence.

@ line 57: replace “–CF3” by “trifluoromethyl (-CF3)”

Replaced accordingly.

@ line 62: reference 62 is not appropriate as this paper does not deal with Trifluoromethyl substituted phenyls

In reference 22, trifluoromethyl phenyl substitution on the hydrazone moiety showed the potent activity. 

@ section 2: This is like “Materials and Methods” and should appear later in the manuscript. Please have a close look at the “Research Manuscript Sections” (https://www.mdpi.com/journal/molecules/instructions#preparation)

We moved the section at the end of the manuscript.

@ spectroscopic data: Please provide assignment of the most obvious IR bands and assignment of the protons of the 1H NMR spectrum for all compounds.

We assigned the bands and peaks for compound 1. It is very difficult to assign the NMR peak for these compounds without 2D spectra. 

@ line 71: better say: 1H and 13C NMR spectroscopy, @ line 72: …NMR spectra

We added as suggested by the reviewer.

@ line 100: ??

Added "Compounds".

The paragraph lines 101-107 should appear straight after describing the procedure (after line 89).

We have combined the procedures and characterization data together.

@ line 102/109/117/126/…: Bis (etc.) should be with capital letter.

We capitalized the first letter of each compound.

@ “technique suggested by the Clinical and Laboratory Standards Institute (CLSI)” References 29 and 30 do not refer to CLSI but are paper of the research group. Please exchange the citation for appropriate ones.

We cited accordingly.

@ line 444: “MBEC was the well…” It was rather the concentration in the well.

We don't think that we need to make any modification in this.

@ line 451: There are also redundant self-citations which do not describe the synthesis itself but its application.

Yes, citations describe the synthetic protocol of pyrazole derivatives.

@ line 455: “comprise” is a strange vocabulary: influence, impact, …

We changed it to impact.

@ line 460: “…but no activity against the Gram-negative strains” Please provide reasons for the missing activity of the compounds against Gram-negative bacteria. Maybe this can be related to the mechanism of antibacterial action. The author do not provide any insight into the mode of action. This is important to overcome resistance and design prper drug candidates. The authors should provide studies on the mode of action in the manuscript.

We are determining the mode of action by using the genomics and proteomic analysis of the treated bacteria. These results will be reported somewhere else.

@ Table 1: The size of the phenyl residues is different. Please unify.

We compressed the size of the phenyl residue.

@ Table 2: “Antimicrobial studies novel compounds” please revise this sentence. Explain the abbreviations V and D, and what the data mean, probably MIC values in µg/ml as determined by…assay. Please add these information to the caption. How many repeats were performed, n=? #6 appears two time in the table, #7 is missing: is this a typo?

We appreciate the reviewer for critically analyzing Table 2. We added the missing information.

@ line 497 and in the Supplementary Material: The use of “&” is not common in scientific papers; also @ line 536: “To our delight” is not scientific.

We changed "&" to "and" and removed "To our delight."

@ line 498 “several compounds”: please specify.

We changed "several" to "Most of the"

@ line 503: The authors discuss lipophilicity. Please provide data, e.g. logP value of the compounds.

Physicochemical properties and mode of action will be reported somewhere else.

@ Figure 2: n=?

n = 3, added in Figure 2.

@ Figure 3: It is hard to read due to usage of the same colors (green, orange) for two different compounds (although the color of the dots differ). Why was DMSO assessed after 12 and 24 h?

We didn't assess DMSO after 10 h. 

@ line 534: a number of efforts are being made ?

a number of efforts are being made to eliminate MRSA persisters by using novel antibiotics 

@ line 536: the four potent: Name the compounds using their numbers.

We added the numbers.

@ Figure 4: It is not necessary to insert differently colored dots to the prahs. This just confuses the reader. Use one color per compound.

Different compounds have different colors.

@ line 562: This was already mentioned in line 34.

Thank you for pointing out this. We removed this line.

@ line 568: the abbreviation was already explained in line 432.

Corrected accordingly.

@ line 581: “Nevertheless, these potent compounds can be modified to increase their pharmacological properties and reduce their toxicity profile” This is an empty phrase as it is true for any drug.

We are working in this direction.

The interpretation of the biological data (MIC) with respect to the chemical structure of the derivatives and the adducing of SAR is confusing. Please clarify this section and provide clear statements which chemical modifications are valuable for further drug optimization/design. What is the difference between (a) the halogen substituents, and (b) their different location at the phenyl residue?

We added a statement of meta and para isomers. For other isomers, we did not find a clear relationship between positional isomers.

The titel of the manuscript states “drug resistant bacteria”. I am wondering, since the authors did not only investigate resistant strains. Moreover, there is no information about the particular resistance of the used strains. Please provide information in the manuscript.

Five of the 10 tested bacteria are antibiotic-resistant.

Although there will be a close editing by the publisher, the authors are kindly ask to correct some formal errors and inconsistency, such as (the following list is to be understood exemplarily): different style of presenting the authors’ names (@ line 5), typos (@ line 10), explanation of abbreviation (NMR is not explained, whereas FTIR is, please be consistent), @ line 81: 30 mL of…, @ line 91: 0.55 mmol of aniline derivative, remove abundant spaces in the nomenclature names (@ lines 102, 109) in the numeration (@ line 145, 146), use of space e.g., @ line 110, 198, 253, 254, 273, 274,… 378,…; different fonts e.g., @ line 295, 305, 314, 324,…, 366-373 ; @ line 389 vs. 396, 397 usage of capital letters?, @403: HEK-293; @ line 408 capital letters Time Kill Assay?, @ line 414: 2 h; @ line 417: using dash/hyphen time-kill and before without, also 96-well, e.g. lines 434/437/438/439;

We really appreciate the reviewer for these comments. We proofread the manuscript carefully and highlighted the changes.

Reviewer 3 Report

Drug-resistant bacteria present a dramatic threat to public health, the discovery of antibiotics to overcome resistant bacteria strains is warranted.  As a continuous study to develop pyrazole-based derivatives as growth inhibitors of drug-resistant bacteria, the authors describe, in this submitted manuscript titled “Synthesis of 3,5-Bis(trifluoromethyl)phenyl-Substituted Pyrazole Derivatives as Potent Growth Inhibitors of Drug-Resistant Bacteria”, the synthesis of thirty novel pyrazole derivatives through a previously reported synthetic method, as well as their SAR against Gram-positive bacterial strains and biological effect against MRSA persisters. Overall, this manuscript identified several potent structures showing low MIC, potent activity against MRSA persisters, and antibiofilm properties.

I find this submitted manuscript intriguing and insightful for future medicinal chemistry programs of developing related chemical scaffolds as antimicrobial agents. The submitted manuscript is well written and organized. Compounds synthesized are well characterized. I strongly recommend the publication of this manuscript in Molecules. However, prior to its publication, the authors are recommended to address the following minor points:

Minor points:

  • Many compounds reported in this manuscript showed great potency and abilities to overcome MRSA persisters. It is nevertheless unfortunate that these structures seem to have very poor safety profiles as demonstrated in the cytotoxicity assay. Can the authors please briefly discuss potential SAR or other solutions to avoid this?
  • In the General consideration, please provide information for HRMS instrument.
  • In the characterization data of compound III (page3), ppm error of HRMS is too big (~18 ppm).
  • On line 464: “Isopropyl aniline derivative (3) inhibited the growth of S. aureus strains with MIC values”, it should be Isopropyl aniline derivative (2).
  • In the Table 2: there are two compound “6”, please correct this.
  • In the supplementary: page 2: “4-[3-[3,5-bis(trifluoromethyl) phenyl]-4-formyl-pyrazol-1-yl] benzoic acid (1).”, it should be (III).

Author Response

  • Many compounds reported in this manuscript showed great potency and abilities to overcome MRSA persisters. It is nevertheless unfortunate that these structures seem to have very poor safety profiles as demonstrated in the cytotoxicity assay. Can the authors please briefly discuss potential SAR or other solutions to avoid this?

We have discussed the SAR for antibacterial properties. Unexpected cytotoxicity may be due to the lipophilicity of the compounds.

  • In the General consideration, please provide information for HRMS instrument.

Thank you for this useful comment, we added the HRMS instrument information.

  • In the characterization data of compound III (page3), ppm error of HRMS is too big (~18 ppm).

NMR and FTIR data support the structure of the compound.

  • On line 464: “Isopropyl aniline derivative (3) inhibited the growth of S. aureus strains with MIC values”, it should be Isopropyl aniline derivative (2).

We appreciate the reviewer for pointing out this mistake. We corrected it.

  • In the Table 2: there are two compound “6”, please correct this.

We corrected it.

  • In the supplementary: page 2: “4-[3-[3,5-bis(trifluoromethyl) phenyl]-4-formyl-pyrazol-1-yl] benzoic acid (1).”, it should be (III).

Thank you!

We corrected it.

Round 2

Reviewer 2 Report

The authors did a diligent job to address the concerns reported before. However, there are still some aspects I wish to mention.

The size of some phenyl residues is still different (e.g., 19 and 20, 14 and 29) although the authors state that they compressed the size of the phenyl residues. Well, this is just a visual aspect and does not impair the scientific value of the manuscript.

The authors added information to the caption of Table 2. Anyways, the unit of the MIC value is still missing. Please mention that the data are given as “µg/ml”.

I suggest that Figure 3 is still subject to improvement. Using two orange curves with blue dots (V and DMSO) is just confusing. I am aware that the dots slightly differ, but that is hard to see. Again, using two green curves (28 and 29) is also not very beneficial – especially when looking at the curves between 4 h and 7 h.

Probably I missed this point in the revised manuscript, but did the authors react to the following concern, as mentioned before: “Moreover, there is no information about the particular resistance of the used strains. Please provide information in the manuscript.” In the caption of Table 2, the authors term it as “antibiotic-resistant”. However, I did not find a clear assignment to the particular antibiotics which the used strains are resistant to.

The authors did not adjust the changes of the affiliation in the supplementary material (United StateS). Moreover, there are still “&” in the captions and the nomenclature names are not capitalized. This contradicts the statements made by the authors: “We changed "&" to "and"” as well as “We capitalized the first letter of each compound.” I suggest to change this seriously.

I consent if the authors wish to publish their results of the studies on the mechanism of action in an upcoming paper. Nevertheless, I think reporting on logP values to describe the lipophilicity of the compounds should be done in the current paper, since the authors adduce lipophilicity to explain different MIC of the compounds.

Author Response

We thank the reviewer for pointing out the mistake to make our manuscript better. We hope that we addressed all the points raised by the reviewer.

The size of some phenyl residues is still different (e.g., 19 and 20, 14 and 29) although the authors state that they compressed the size of the phenyl residues. Well, this is just a visual aspect and does not impair the scientific value of the manuscript.

We tried our best to make the same size of all the structures. Due to different substituents, their sizes look different.

The authors added information to the caption of Table 2. Anyways, the unit of the MIC value is still missing. Please mention that the data are given as “µg/ml”.

Thank you for pointing this out. We added "µg/mL".

I suggest that Figure 3 is still subject to improvement. Using two orange curves with blue dots (V and DMSO) is just confusing. I am aware that the dots slightly differ, but that is hard to see. Again, using two green curves (28 and 29) is also not very beneficial – especially when looking at the curves between 4 h and 7 h.

We changed the figure.

Probably I missed this point in the revised manuscript, but did the authors react to the following concern, as mentioned before: “Moreover, there is no information about the particular resistance of the used strains. Please provide information in the manuscript.” In the caption of Table 2, the authors term it as “antibiotic-resistant”. However, I did not find a clear assignment to the particular antibiotics to which the used strains are resistant to.

We added "methicillin-resistant and vancomycin-resistant (VRE)" in Table 2.

The authors did not adjust the changes of the affiliation in the supplementary material (United StateS). Moreover, there are still “&” in the captions and the nomenclature names are not capitalized. This contradicts the statements made by the authors: “We changed "&" to "and"” as well as “We capitalized the first letter of each compound.” I suggest to change this seriously.

We thank the reviewer for finding this mistake. We corrected the supporting information as well.

I consent if the authors wish to publish their results of the studies on the mechanism of action in an upcoming paper. Nevertheless, I think reporting on logP values to describe the lipophilicity of the compounds should be done in the current paper, since the authors adduce lipophilicity to explain different MIC of the compounds.

We added the clog P values in the discussion.